# Detection of Trace Elements/Isotopes in Olympic Dam Copper Concentrates by nanoSIMS

**Mark Rollog [1,\*]** , **Nigel J. Cook [1]** , **Paul Guagliardo [2]** , **Kathy Ehrig [3]** , **Cristiana L. Ciobanu [1]** and **Matt Kilburn [2]**

[1] School of Chemical Engineering, The University of Adelaide, Adelaide, 5005 SA, Australia; nigel.cook@adelaide.edu.au (N.J.C.); cristiana.ciobanu@adelaide.edu.au (C.L.C.)
[2] Centre for Microscopy, Characterisation and Analysis, The University of Western Australia, 35 Stirling Highway, Crawley, 6009 WA, Australia; paul.guagliardo@uwa.edu.au (P.G.); matt.kilburn@uwa.edu.au (M.K.)
[3] BHP Olympic Dam, 55 Grenfell St., Adelaide, 5000 SA, Australia; kathy.ehrig@bhp.com
* Correspondence: mark.rollog@adelaide.edu.au; Tel.: +61-466-848-699

**Abstract:** Many analytical techniques for trace element analysis are available to the geochemist and geometallurgist to understand and, ideally, quantify the distribution of trace and minor components in a mineral deposit. Bulk trace element data are useful, but do not provide information regarding specific host minerals—or lack thereof, in cases of surface adherence or fracture fill—for each element. The CAMECA nanoscale secondary ion mass spectrometer (nanoSIMS) 50 and 50L instruments feature ultra-low minimum detection limits (to parts-per-billion) and sub-micron spatial resolution, a combination not found in any other analytical platform. Using ore and copper concentrate samples from the Olympic Dam mining-processing operation, South Australia, we demonstrate the application of nanoSIMS to understand the mineralogical distribution of potential by-product and detrimental elements. Results show previously undetected mineral host assemblages and elemental associations, providing geochemists with insight into mineral formation and elemental remobilization—and metallurgists with critical information necessary for optimizing ore processing techniques. Gold and Te may be seen associated with brannerite, and Ag prefers chalcocite over bornite. Rare earth elements may be found in trace quantities in fluorapatite and fluorite, which may report to final concentrates as entrained liberated or gangue-sulfide composite particles. Selenium, As, and Te reside in sulfides, commonly in association with Pb, Bi, Ag, and Au. Radionuclide daughters of the $^{238}$U decay chain may be located using nanoSIMS, providing critical information on these trace components that is unavailable using other microanalytical techniques. These radionuclides are observed in many minerals but seem particularly enriched in uranium minerals, some phosphates and sulfates, and within high surface area minerals. The nanoSIMS has proven a valuable tool in determining the spatial distribution of trace elements and isotopes in fine-grained copper ore, providing researchers with crucial evidence needed to answer questions of ore formation, ore alteration, and ore processing.

**Keywords:** radionuclides; trace elements; isotopes; nanoSIMS; copper concentrate; Olympic Dam

## 1. Introduction

Systematic analysis of trace elements (or isotopes) in ore samples is beneficial on multiple fronts. Valuable geochemical insight may be gained by detailed analysis of ore deposits at every scale, from regional trends useful for mineral exploration down to the nanoscale deportment of trace elements. The association of certain trace elemental or isotopic components in a mineralized system may aid

in determining ore genesis timelines, mineralization conditions, source rocks, and subsequent alteration via interaction with hydrothermal fluids or metamorphism—just to name a few. Trace element/isotope analyses may also provide economic advantages. Many mining operations employ complex processing flowsheets and monitoring strategies which take into account variable ore compositions, particle sizing, wash water chemistry, acid or base leach compositions and concentrations, oxidation/reduction potentials, and temperature—all optimized through cost-benefit analysis algorithms. Detailed monitoring of each component of the ore, not just the economic ones, may eventuate into lowering the unit costs of production. To this end, measuring concentrations of trace elements *not* currently recovered, and determining their mineralogical distributions, has become routine for many mine operations. Although a comprehensive review of trace element methods and applications is far beyond the scope of this study, our intention is simply to emphasize the importance of such analyses and to demonstrate the utility of an alternative analytical technique which offers significant advantages over other instrumentation.

To process engineers, trace components nominally fall into one of three categories—benign, beneficial or deleterious. Benign elements are of little economic interest and do not, in low concentrations, affect the recovery of economic elements. Beneficial components may either be elements not currently marketable in quantities available, but which may be in the future (notably certain 'strategic' elements increasingly applied in high-tech applications such as rare earth elements (REE), Ni, Co, Te, Sc, and Nb); or elements which are currently recovered through secondary processes (e.g., Au, Ag, and platinum group elements from electrorefining slimes, V from coal/oil production, Re from molybdenum ores, or Ga, Ge, and In from zinc production). The provenance or deportment of these elements in the deposit may be poorly or incompletely understood. Deleterious components are generally either penalty elements (e.g., Pb, Bi, Sb, or As in copper concentrate [1]), elements which cause issues with processing (e.g., halogens), or environmentally sensitive elements (e.g., Hg).

Regardless of its beneficial/deleterious/benign classification, a robust understanding of the behavior, as well as concentration, of the component in question is critical knowledge to the geologist, mineralogist, and metallurgist alike. Bulk analyses of trace elements (and isotopes) are available via many well-established methods [2], but crucial information may be lost when measuring homogenized aliquots. In situ analysis of specific minerals eliminates this uncertainty to some extent but may be difficult at the resolution and minimum detection limits required for certain elements or isotopes. Few currently available instruments provide the resolution and sensitivity necessary for these in situ analyses.

One such analytical platform is nanoscale secondary ion mass spectrometry (nanoSIMS) manufactured by CAMECA. As with other instruments, it has both positive and negative characteristics [3]. Drawbacks include expense, rarity (currently ~45 worldwide), isobaric mass interferences (inherent in all mass spectrometry), and the inability to quantify isotope or element concentrations in complex matrices—including most minerals. Advantages include multicollection capabilities (seven detectors on the 50L instrument at the Centre for Microscopy, Characterisation, and Analysis (CMCA) in Perth, Western Australia); interchangeable ion sources ($Cs^+$ for organics and anions, $O^-$ for most cations); mass resolution to 0.1 atomic mass unit (amu); low minimum detection limits; and excellent spatial resolution (ultimately to 40 nm). The pairing of these last two points is arguably its greatest advantage over other instruments.

NanoSIMS has provided valuable insight in such diverse fields as cell metabolism [4,5], cosmochemistry (e.g., Reference [6]), metallurgy (e.g., Reference [7]), nanoelectronics (e.g., Reference [8]), and geosciences [3]). In ore deposit research, nanoSIMS mapping has proven invaluable for unraveling the internal compositional heterogeneity of ore minerals, notably with respect to extending understanding of the location and speciation of Au in pyrite and arsenopyrite and how sulfide minerals form in nature (e.g., References [9–11], and not least, to provide critical constraints on interpretation of geochronological data [12]. Although generally not quantifiable for trace elements—at least in complex mineral assemblages—isotope ratio analyses are robust. Applications include $^{34}S/^{32}S$ in sulfides and

sulfates (e.g., References [13,14], $^{18}O/^{16}O$ in meteorites (e.g., Reference [15]), $^{13}C/^{12}C$ and $^{15}N/^{14}N$ in biological samples (e.g., Reference [16]), and $^{207}Pb/^{206}Pb$ dating of zircon and baddeleyite [17].

In this contribution, we document how nanoSIMS mapping represents a scale-appropriate analytical technique that can provide an advanced understanding of the physical state of trace elements of interest in complex ores and their processing materials. Drawing on selected examples from the Olympic Dam (OD) mining-processing operation, South Australia, we discuss the utility of nanoSIMS mapping for investigation of trace element and radionuclide (isotope) distributions and how correlations between different elements relate to mineralogy. Although only about 20 elements were mapped for this study, the nanoSIMS platform is capable of in situ analysis of virtually the entire periodic table at a resolution well-suited to complex, fine-grained ores and processing materials generated from them. NanoSIMS capabilities used together with complementary microanalytical techniques represent an important contribution towards the development of semi-quantitative deportment models for trace components of the ore. These are, in turn, a necessary pre-requisite for future extraction of potential by-products and for efforts to reduce or eliminate deleterious components in final concentrates.

## 2. Background

The Olympic Cu-Au province is located along the eastern margin of the Gawler Craton, South Australia [18,19] and hosts numerous Cu-Au-(U) deposits of iron oxide copper gold (IOCG) type, in which hematite is the most abundant mineral. The Olympic Dam Cu-U-Au-Ag deposit is by far the largest of these. Olympic Dam ores are mineralogically complex, characteristically fine-grained and feature mineral intergrowths down to the sub-micron-scale [20,21]. Although the deposit is exploited for Cu, U, Au, and Ag, it is also enriched in many other elements, including REE, Y, W, Mo, Sn, As, Co, Nb, Se, Te, F, Ba, and Sr [20]. Although the complexity of Olympic Dam ores was recognized early on [22,23], research over the past decade has been able to take advantage of the array of modern microanalytical instrumentation to systematically document the mineralogy of the deposit and element distributions in processing streams. These studies have used scanning electron microscopy (SEM), electron probe microanalysis (EPMA), laser ablation inductively coupled plasma mass spectrometry (LA-ICP-MS), focused ion beam SEM (FIB-SEM) and transmission electron microscopy (TEM), often on the same sample material. Outcomes have emphasized the fine-scale intergrowths and also relationships between ore and gangue minerals, the latter also playing a significant role in concentrating trace elements of interest or concern.

Ores and their component minerals display evidence of multiple cycles of in situ replacement and phase transformation but also remobilization and recrystallization (e.g., [20,24–28]. Copper concentrates produced by froth flotation contain ca. 80% Cu-(Fe)-sulfides but are also enriched in many minor and trace metals (Ag, As, Bi, Au, Co, etc.) relative to ore feed; in many cases, those trace elements mostly occur in solid solution in the host sulfides or as included grains of discrete minerals. Uranium is recovered by sulfuric acid leaching of both flotation tailings and copper concentrates.

One critical aspect of the overarching approach in the present study is the characterization of distinct distributions and behaviors of some elements/isotopes/minerals in samples taken from different stages of mineral processing. The sample suite encompasses a few drill core specimens but is largely focused on flotation concentrates (FC) and concentrate leach discharge (CLD) samples. Drill core samples provide a snapshot into the naturally formed deposit. Flotation concentrates are produced via ore comminution followed by froth flotation, upgrading sulfides and potentially remobilizing nanoparticulates. Concentrate leach discharges are flotation concentrates which have undergone sulfuric acid leaching; extensively altering mineralogy on the nano-, micro-, and macroscales and redistributing trace elements and isotopes at anthropogenic conditions. A simplified flow sheet of ore processing at Olympic Dam may be found in Schmandt et al. [29]. This study presents just a few examples of beneficial trace element/isotope analyses, including examples of both geochemical and economic interest.

Chalcopyrite, bornite and chalcocite are the predominant copper minerals in Olympic Dam ore [20,28]. These sulfides are commonly intergrown with one another (notably bornite-chalcocite symplectites) and with U- and REE-bearing phases. Uranium-bearing minerals are uraninite, coffinite and brannerite [24–27], with significant concentrations also present in solid solution and as nanoparticles within hematite [30–32]. Gangue minerals are dominantly hematite, sericite, quartz, feldspars, pyrite, siderite, chlorite, baryte and several dozen other minor and trace minerals [20].

Rare earth elements are abundant and ubiquitous within the Olympic Dam ore [20] but are not recovered. The two major REE minerals at Olympic Dam are bastnäsite, $REE(CO_3)F$ [33] and florencite, $REEAl_3(PO_4)_2(OH)_6$ [34]; these plus minor xenotime ($YPO_4$), synchysite [$CaREE(CO_3)_2F$], monazite [$REE(PO_4)$], and parisite [$CaREE_2(CO_3)_3F_2$] comprise ~0.2 wt.% of the ore [20]. Additionally, woodhouseite-svanbergite group minerals, fluorapatite, zircon, fluorite, and uranium/thorium minerals are known to host low quantities of REE [35–37].

Precious metals within the Olympic Dam resource grade at an average of 0.30 g/t Au and 1.3 g/t Ag [38]. Assays of various processing stream concentrates show platinum group elements to be insignificant (<3 ppb). Most of the gold at Olympic Dam is either present in the native form or electrum [20] and is recovered efficiently from refinery slimes after electrorefining. Silver at Olympic Dam is rarely native and is either incorporated within Cu-(Fe)-sulfides or as discrete minerals combined with Pb, Bi, Se, and/or Te [20].

Uranium-bearing copper ore from iron oxide-copper-gold deposits such as Olympic Dam requires the additional consideration of daughter radionuclides (RNs) from $^{238}U$, $^{235}U$, and $^{232}Th$ decay. Most of these daughters have either very long ($^{234}U$ = 245,500 years) or very short ($^{214}Po$ = 164 μs) half-lives and therefore have either specific activities or concentrations that are too low to be of any concern. There are, however, three radionuclides which require routine monitoring due to their combination of measurable concentrations and high specific activities: $^{226}Ra$ (1601 years), $^{210}Pb$ (22.3 years), and $^{210}Po$ (138.3 days). These radionuclides constitute most of the activity in Olympic Dam copper concentrates [39].

## 3. Methods

The strategy for selecting samples for analysis was based on building a broad database of isotope distribution maps containing data for as many Olympic Dam minerals as possible. Priority was given to minerals predicted to host radionuclides, such as U-, REE-, and Pb-bearing phases, and baryte. Most ore minerals (e.g., chalcopyrite, bornite, chalcocite) and gangue minerals (e.g., quartz, hematite, white mica (e.g. sericite), fluorite, carbonates) were not targeted as such, but the finely-intergrown nature of the ore guaranteed that prevalent minerals were peripherally sampled many times each. For example, individual 50 μm-diameter grains may contain 10+ unique mineral phases. The images presented here represent typical examples for the topics discussed, selected from over 3200 isotope maps collected from 205 sample grains.

Samples were analyzed as described in Rollog et al. [40]. Briefly, copper concentrates from Olympic Dam were mounted in 25 mm-diameter epoxy resin rounds which were then polished. Both FC and CLD samples were prepared. Drill core pieces were mounted in epoxy resin, ground flat, and polished. Mounts were surveyed using an FEI Quanta450 field emission gun scanning electron microscope (FEG-SEM) equipped with a backscatter secondary electron (BSE) detector and an EDAX electron dispersive spectroscopy (EDS) detector, located at Adelaide Microscopy, The University of Adelaide. Grains of interest were located, imaged, characterized for mineralogy, and select grains were mapped for elemental distributions.

Isotopic maps were produced on CAMECA 50 and 50L nanoSIMS instruments (CAMECA, Gennevilliers, France) at the Centre for Microscopy, Characterisation, and Analysis, located at The University of Western Australia, Perth, Western Australia. Both the nanoSIMS 50 (5 detectors, $Cs^+$ ion source) and nanoSIMS 50L (7 detectors, $O^-$ ion source) were used. For analyses on the nanoSIMS 50L, the seven detectors were carefully tuned to the desired isotopes and grains of interest were presputtered, then mapped (50 × 50 μm raster area, 50 pA ion current, D1 = 2, ES = 2, AS = 0, 512 × 512 pixels (px),

3 planes, 5 ms/px, effective beam diameter ≈ 400 nm). Following this round of mapping, the instrument was retuned to six different isotopes and one redundant isotope, usually $^{54}$Fe (for comparison and alignment between runs), and the grains were mapped a second time (50 × 50 μm raster area, 250 pA ion current, D1 = 2, ES = 2, AS = 0, 512 × 512 px, 5 or 6 planes, 5 ms/px, effective beam diameter ≈ 700 nm). A similar procedure was used for the nanoSIMS 50, although most grains were mapped only once (50 x 50 μm raster area, 2 pA ion current, D1 = 2, ES = 2, AS = 2, 512 × 512 px, 1-3 planes, 15 ms/px, egun on, effective beam diameter ≈ 400 nm). There are two exceptions to the above; maps of samples 01FC58 and 01FC13 shown as Figures 2 and 3 were mapped at a resolution of 256 × 256 px.

Images presented in this contribution represent samples from six distinct batches of copper concentrate, mounted independently (Table 1). No significant differences in mineralogy were reported between collection dates, per sample type. NanoSIMS images were collected over three visits to CMCA in Perth during 2017.

**Table 1.** The particles (sample grains) analyzed by nanoSIMS, per concentrate type and collection date. Between 7 and 26 isotope maps were collected for each, averaging ~14. A selected subset of these analyses is presented in this study.

| Sample Type | Collection Date | | | |
|---|---|---|---|---|
| | 08/2015 | 11/2016 | 12/2016 | 12/2017 |
| Flotation concentrate (FC) | 13 | | 43 | 16 |
| Acid-leached concentrate (CLD) | 51 | | 31 | 44 |
| Drill core samples | | 7 | | |

Images were processed with ImageJ [41,42] and the OpenMIMS plugin [43]. Dead-time corrections were applied, and multiple image-planes were summed. Occasionally, second runs did not overlap perfectly with initial runs. Images were cropped accordingly to remove non-overlapping regions. Color composite images of 3 or 4 major isotopes were generated to more easily associate faint, nebulous trace element patterns with grain-scale features in the BSE images.

To verify what was seen in nanoSIMS maps, LA-ICP-MS spot analyses were performed on selected samples using an ASI RESOlution-LR ArF excimer laser ablation system (Adelaide Microscopy, The University of Adelaide) with a large format S155 sample chamber (Laurin Technic Inc.) coupled to an Agilent 7900× ICP-MS. Instrument specifics include a 13 μm spot diameter, fluence 3.5 Jcm$^{-2}$, and repetition rate of 10 Hz. Counts-per-second (isotopic) were converted to counts-per-second (elemental) using global isotopic abundances [44]. RN concentrations were not quantifiable due to a lack of radionuclide standards so the traces presented are for comparison only. In this study, however, the *lack* of signal is a significant result.

## 4. Results and Discussion

Since they are easily detectable in bulk samples, trace elements can be tracked (and in most cases recovered or removed, if desired) during processing, but details of their micro- or nanoscale deportment are not easily discerned by low-resolution techniques. Close inspection of nanoSIMS maps revealed a number of associations that may be largely undetectable using other methods due to extremely low concentrations and/or the scale of mineral intergrowths.

### 4.1. Precious Metals

Although methods of precious metal recovery from electrorefining slimes are well established, they are not necessarily efficient if the mineralogical balance for those elements is inadequately studied. Hidden associations revealed by nanoSIMS mapping may offer pathways to more cost-effective recovery solutions. The presence of Au and Ag can be seen in various sulfide and non-sulfide minerals on

nanoSIMS maps of Olympic Dam samples, likely as nanoscale inclusions. Gold and Ag are recovered only from the copper concentrates, not flotation tailings where the Au-Ag grades are sub-economic.

Gold concentrations average approximately 10 ppm in copper concentrates. Although most Au at the Olympic Dam is found as micron-sized native gold or electrum, there is evidence of remobilization of Au throughout the deposit and at a range of scales. Hydrothermal fluids containing many (likely complexed) cations have circulated through the deposit multiple times [20], depositing their metal load when conditions are appropriate. One mechanism for brannerite formation involves U-rich fluids interacting with rutile [27]. It is not unreasonable to think that the U-rich fluids may contain other elements, such as Au and Te, which precipitate simultaneously as nanoparticles (Figure 1). The ability of brannerite to incorporate Au and Te has been observed in Buryatian placer nuggets up to 50 g in weight [45].

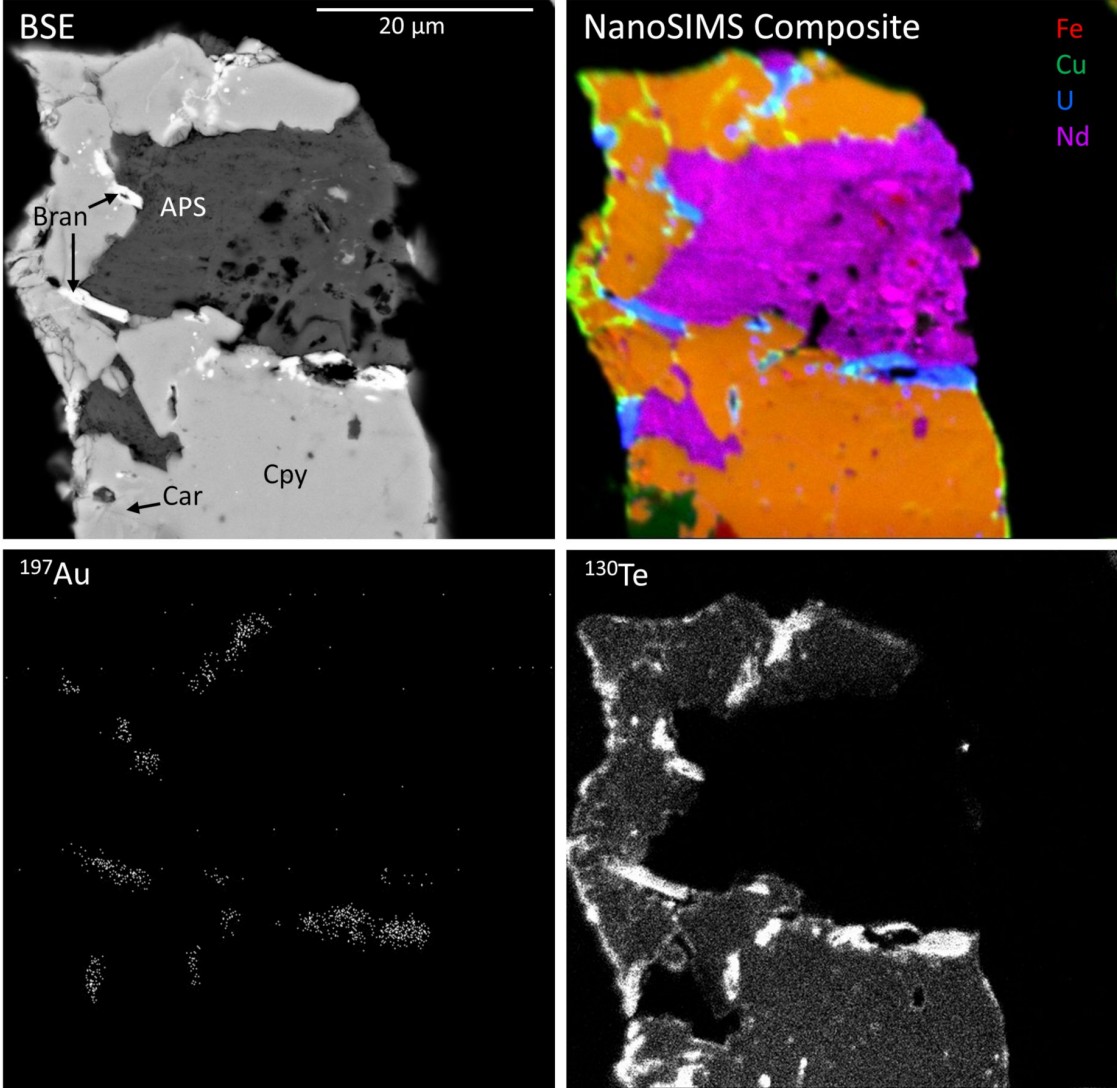

**Figure 1.** The backscatter secondary electron detector (BSE) image and nanoSIMS images of grain 05CLD27. The composite image contains nanoSIMS overlays of Fe (red), Cu (green), U (blue), and Nd (magenta). Note that orange is the result of a red and green overlap (chalcopyrite in this sample). Abbreviations: (Bran) = brannerite; (Cpy) = chalcopyrite; (Car) = carrollite; (APS) = aluminum phosphate-sulfate phase. Trace levels of Au can be seen coincident with Te, likely as cogenetic nanoparticles in brannerite.

Silver grades average 1.3 ppm in the Olympic Dam resource [38] but can upgrade to >80 ppm during flotation. It is predominantly found as hessite ($Ag_2Te$) inclusions in sulfides or at concentrations of tens to hundreds of ppm in bornite and chalcocite [20]. Two samples were selected for analyses to determine the distribution and associations of Ag in copper sulfides. The first sample is comprised of coarsely intergrown bornite-chalcocite (Bn-Cc) with a uraninite-dolomite vein fill (Figure 2). The nanoSIMS composite image reproduces the BSE image, but additionally reveals high levels of heavy rare earth elements (HREE) in the uraninite. The Ag is concentrated in chalcocite, but to a lesser extent between the uraninite-dolomite veins. Highly enriched zones are found where chalcocite borders the grain edge. In contrast, Bi favors bornite and is more enriched between the uraninite-dolomite veins. A possible mass interference exists, as $^1H^{208}Pb$ is indistinguishable from $^{209}Bi$ on the nanoSIMS. The example in Figure 3 illustrates that this interference is minor. A survey of multiple lead minerals shows a variable 209 signal, suggesting that this is due to real variable amounts of $^{209}Bi$, which would be expected, and not an inconsistent $^1H^{208}Pb$ signal response. Additionally, scanning mass 209 while focusing the nanoSIMS on a pure Pb standard gives a response four orders of magnitude lower than that of mass 208, suggesting that the $^1H^{208}Pb$ interference is inconsequential.

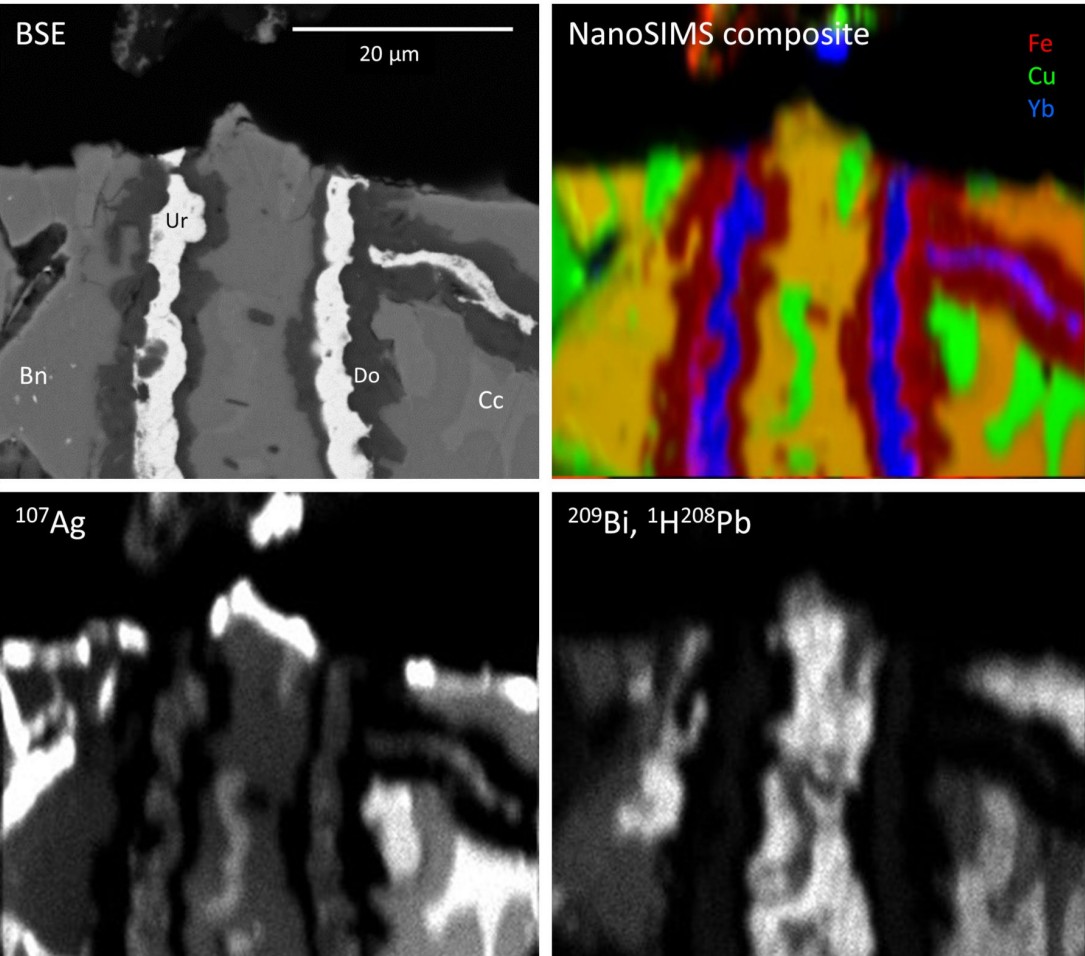

**Figure 2.** The BSE (top left) and nanoSIMS images of sample 01FC58 showing fracture fill uraninite (Ur) and dolomite (Do) within host bornite (Bn) and chalcocite (Cc). The nanoSIMS composite image combines Fe (red), Cu (green), and Yb (blue) maps. Note that orange is the result of a red and green overlap (bornite in this sample).

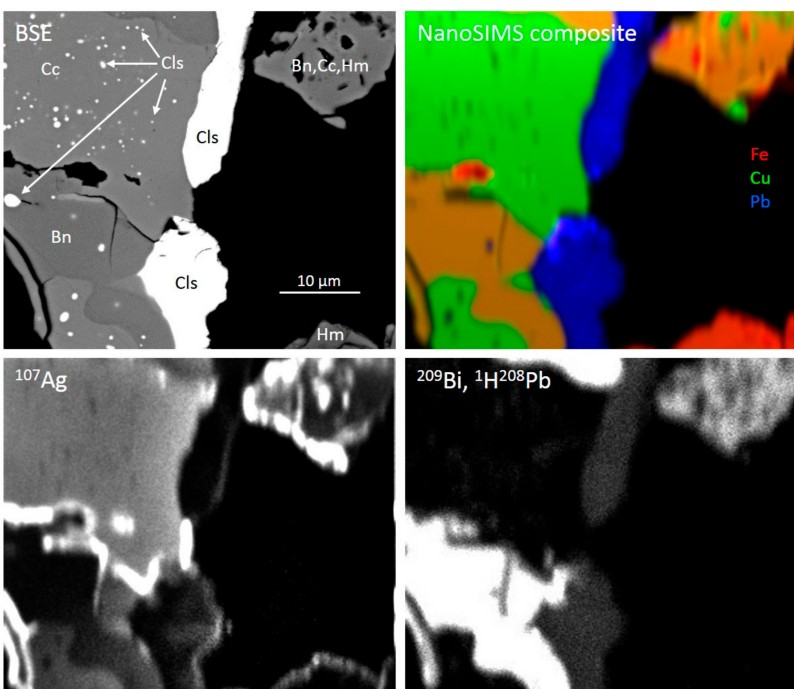

**Figure 3.** The BSE (top left) and nanoSIMS images of sample 01FC13 showing two large clausthalites (Cls) in bornite/chalcocite. Sub-micron clausthalite inclusions are found throughout the chalcocite, and—to a lesser extent—in bornite. The composite image represents overlapping nanoSIMS maps of Fe (red), Cu (green), and Pb (blue). Note that orange is the result of a red and green overlap (bornite in this sample). Silver shows a clear preference for chalcocite whereas Bi is more readily incorporated into bornite. Silver also appears at grain boundaries and in fractures. Isobaric mass interference for mass $^{209}$Bi can barely be seen coinciding with the clausthalite grains, partially caused by $^{1}$H$^{208}$Pb.

The second sample analyzed for Ag distribution consists of coarsely intergrown chalcocite and bornite, both containing sub-micron inclusions of clausthalite (Figure 3). Two large clausthalites are bound to the grain margin. Smaller grains of hematite (lower right) and clausthalite-free bornite with chalcocite and hematite inclusions (upper right) are peripheral to the main grain. Silver is once again concentrated in chalcocite, consistent with Reference [46], with highly enriched zones at Bn-Cc grain boundaries. The peripheral bornite also has elevated Ag, predominantly on the edges and around inclusions. Bismuth distribution mimics that of the first sample, concentrating in bornite. Little to no Bi is found in chalcocite or hematite. The faint signal of mass 209 in the large clausthalites is likely (at least in part) to be the result of mass interference from $^{1}$H$^{208}$Pb, although Bi substitution into Pb minerals is certainly possible [47].

Precious metals at Olympic Dam are well-characterized and recovered via established methods. Gold or Ag associated with copper ore minerals are destined for recovery regardless of their mineralogy, and nanoSIMS maps only serve to verify what is already understood. Trace amounts of Au and Ag can also be seen in various non-ore minerals, but not to levels which are economically inducive to incorporate additional extraction techniques. This information may be useful, however, to geochemists in the form of previously unknown associations or the abundance of nanoinclusions.

### 4.2. Rare Earth Elements

Fluorapatite comprises only 0.1 wt.% of the Olympic Dam deposit [20] but is known to host up to 70 ppm Th and >1.8 wt.% ΣREE [32]. Substitution of REE$^{3+}$ (or Th$^{4+}$) for Ca$^{2+}$ in apatite-group minerals can be extensive. Euhedral fluorapatite grains are frequently found associated with pyrite, as shown in Figure 4. Enriched cores contain primarily light rare earth elements (LREE), although HREE, Ra, and Th also show elevated concentrations. Three of the zoned grains are exposed at

the pyrite grain surface and feature high-Pb zones in the visibly corroded internal grain boundaries. The fourth fluorapatite grain is completely entrained in pyrite and has no corresponding boundary Pb enrichment. Neodymium, Ra, Th, and particularly Tm, also show enrichment at these three grain boundaries, suggesting exposure to REE- and RN-rich solution which did not have access to the entrained crystal. Small aluminum phosphate-sulfate (APS) inclusions exhibit high REE— and moderate RN—enrichment.

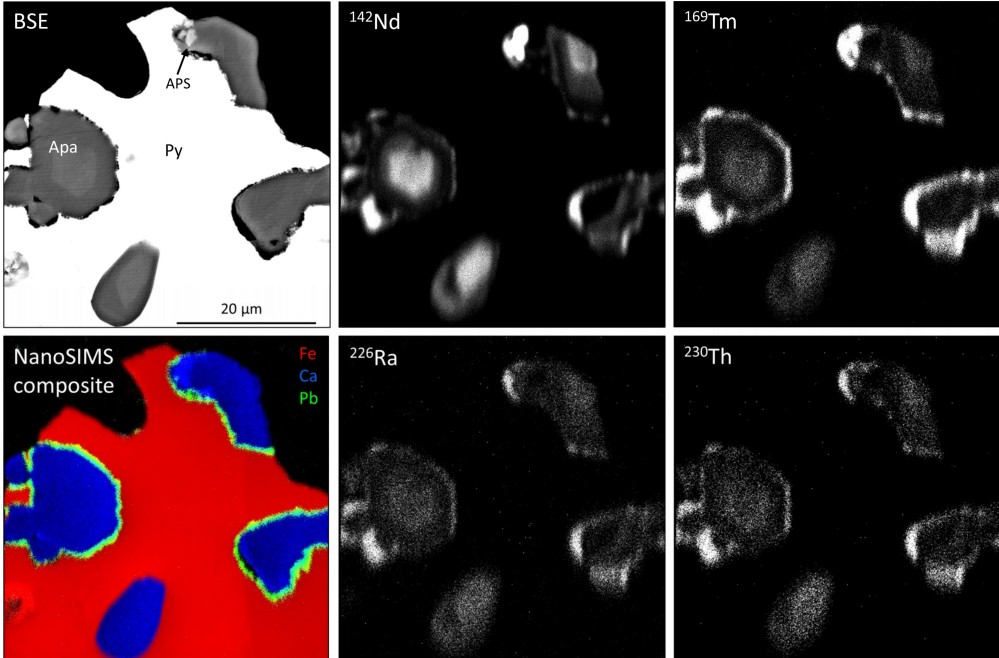

**Figure 4.** The BSE (top left) and nanoSIMS images of sample 10FC41b, showing inclusions of fluorapatite (Apa) within pyrite (Py). The composite image represents overlapping nanoSIMS maps of Fe (red), Ca (blue), and Pb (green). Rare earth elements (represented by $^{142}$Nd and $^{169}$Tm), Ra, and Th are present (and frequently show zoning) in the fluorapatite.

Unzoned fluorapatite associated with sericite-hematite alteration behaves similarly, with even more distinct dissociation between HREE and LREE (Figure 5). The nanoSIMS composite faithfully reproduces the BSE image, with Sr in blue representing fluorapatite (Ca was not mapped). The $^{144}$Nd map highlights enriched areas of LREE not visible in the BSE image, possibly representing florencite, which has roughly the same gray-scale response. Lead is absent from around the apatite and is contained entirely within the chalcopyrite. As in the previous example, the HREE distribution is coincident with RNs.

Fluorite is prevalent at OD, representing approximately1 wt.% of the ore body [20]. Nonetheless, most fluorapatite and fluorite recover to flotation tailings along with the other REE-bearing minerals. Although most of the fluorite that survives froth flotation is removed during sulfuric acid leaching, residual fluorite has the capability of hosting not only elements such as Mg, Sr, and REE, but also Th—at least in concentrations high enough for thermochronology [48]. The comminution process does not always fully liberate gangue minerals, as illustrated by a fluorite grain completely entrained in bornite-chalcocite symplectite (Figure 6). Despite having gone through the sulfuric acid leach tank, this 18 x 25 μm fluorite grain survived. The $Sr^{2+}$ replacement of $Ca^{2+}$ is expected and observed, although there is a Sr-depleted region in the fluorite. The high Sr concentration on the symplectite grain edges is attributed to insoluble $SrSO_4$ formed during the sulfuric acid leach. LREE are also present in trace concentrations, represented by Nd; Tm is absent. Thorium-230 shows a very slight enrichment in the fluorite grain, with additional hot spots correlating to both Nd and Tm—likely due to sub-micron REE-phase (possibly APS-phase) inclusions.

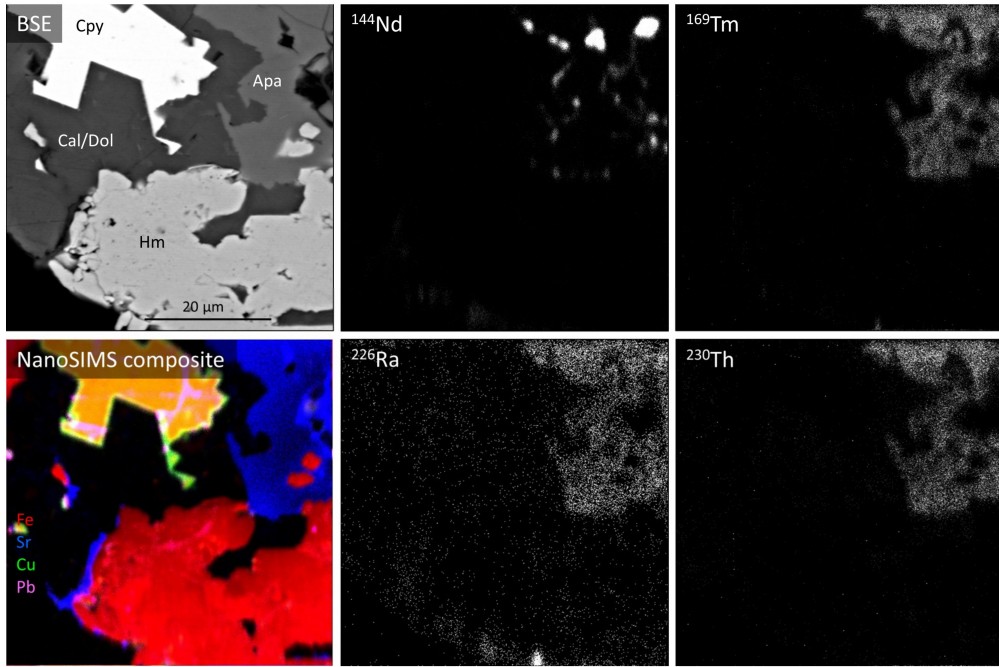

**Figure 5.** The BSE (top left) and nanoSIMS images of sample 10FC68 comprising an assemblage of apatite, calcite/dolomite (Cal/Dol), chalcopyrite, and hematite (Hm). The composite image represents overlapping nanoSIMS maps of Fe (red), Sr (blue), and Cu (green). Note that orange is the result of a red and green overlap (chalcopyrite in this sample). Rare earth elements (represented by [144]Nd and [169]Tm), Ra, and Th are found throughout the apatite (and only the apatite), though no zoning is evident.

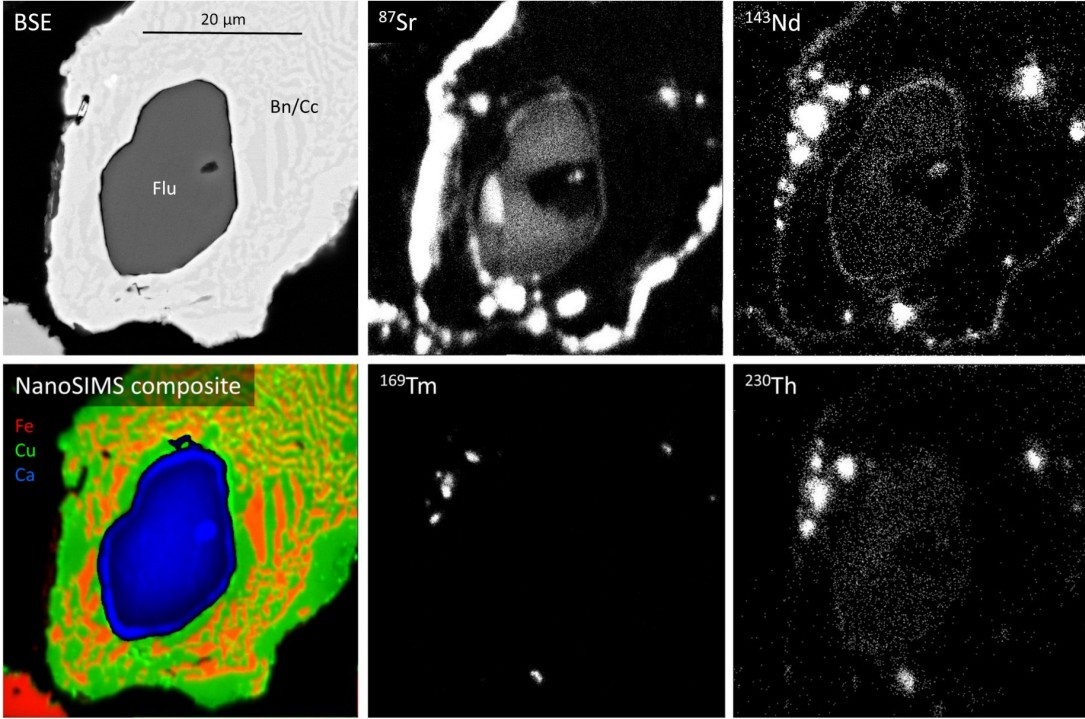

**Figure 6.** The BSE and nanoSIMS images of sample 05CLD86 showing corroded/resorbed fluorite (Flu) within the symplectitic intergrowth of chalcopyrite and bornite. The composite image represents overlapping nanoSIMS maps of Fe (red), Cu (green), and Ca (blue). Note that orange is the result of a red and green overlap (in this case, bornite). $Sr^{2+}$ commonly substitutes for $Ca^{2+}$ in fluorite, frequently resulting in intricate zoning patterns. LREE, represented here by [143]Nd, is associated positively with Sr in the fluorite, but HREE is absent.

Occasionally, a relatively coarse grain of fluorite survives both froth flotation and acid leaching (Figure 7). Compositional zoning with respect to Sr within the crystal is distinct, and correlates to the zoning patterns of Nd and Tm. Thorium-230 is completely absent (not pictured) and $^{206}$Pb is found only as a thin layer of (most likely) insoluble PbSO$_4$ as a result of the sulfuric acid leach process. For verification, a single LA-ICP-MS spot analysis was performed (represented by the dashed yellow circle) and returned La and Ce concentrations above 100 ppm.

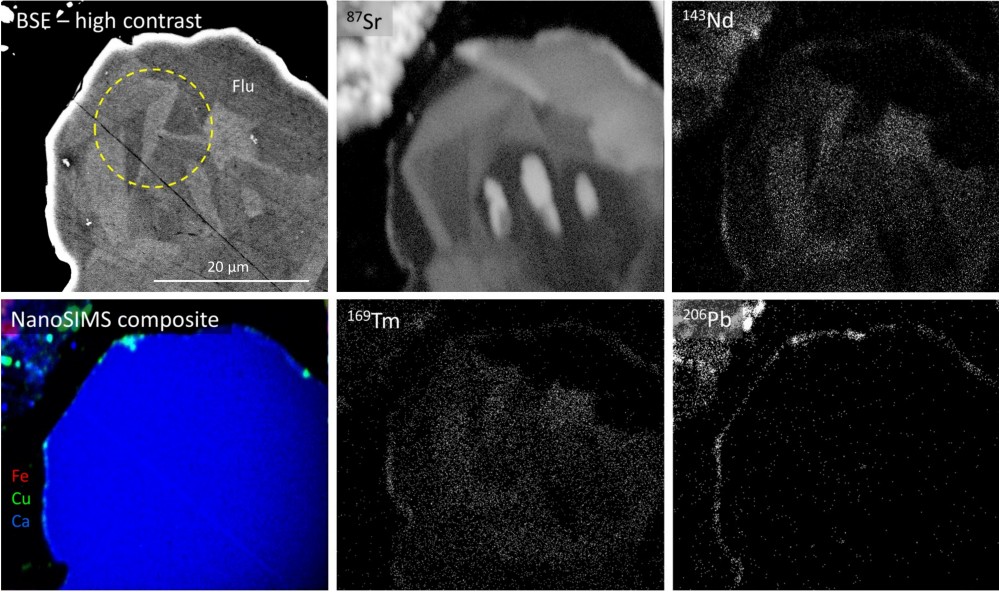

**Figure 7.** The high-contrast BSE (top left) and nanoSIMS images of sample 02CLD39 (coarse fluorite grain). The nanoSIMS composite image represents overlapping nanoSIMS maps of Fe (red), Cu (green), and Ca (blue). Sr$^{2+}$ commonly substitutes for Ca$^{2+}$ in fluorite, frequently resulting in intricate zoning patterns. REE, represented here by $^{143}$Nd and $^{169}$Tm, are associated somewhat negatively with Sr. Lead-206 is found only on the grain surface, likely as insoluble PbSO$_4$. The dashed yellow circle represents the location and relative size of a single 15 μm-diameter LA-ICP-MS spot analysis.

One minor, but important host for REE is the Sr-Ca-dominant aluminum phosphate-sulfates of the woodhouseite-svanbergite group. Detailed investigation of these minerals in OD samples has revealed several distinct members [37]. These phases belong to the alunite supergroup with the general formula [AB$_3$(XO$_4$)$_2$(OH)$_6$], where [A] may be a wide variety of mono-, bi-, or trivalent cations, [B] is almost always Al, and (XO$_4$) may be phosphate, sulfate, or a mixture thereof [49]. At the Olympic Dam, these APS minerals have the A-site dominantly populated by (Sr, Ca, REE), and may contain up to 15 wt.% REE (Figure 8). NanoSIMS maps confirm high levels of Sr, Ba, and Nd, and lesser amounts of Tm, exclusively in the APS phase. This phase is also included by about a dozen small xenotimes, difficult to see in the BSE image but highlighted clearly in the $^{169}$Tm nanoSIMS map. Xenotime is well known to host heavy REE [50,51] but rarely incorporates light REE—as seen in the $^{143}$Nd map. Lead-206 is found throughout both the APS phase and chalcopyrite, although the distribution in APS seems relatively uniform while the host of cracks and fissures in chalcopyrite preferentially accumulate Pb. A single time-resolved laser ablation spot yields high Ca, Sr, La, and Ce, with lesser amounts of Ba and Pb. Thulium is lower in concentration and irregular, suggesting the presence of small inclusions.

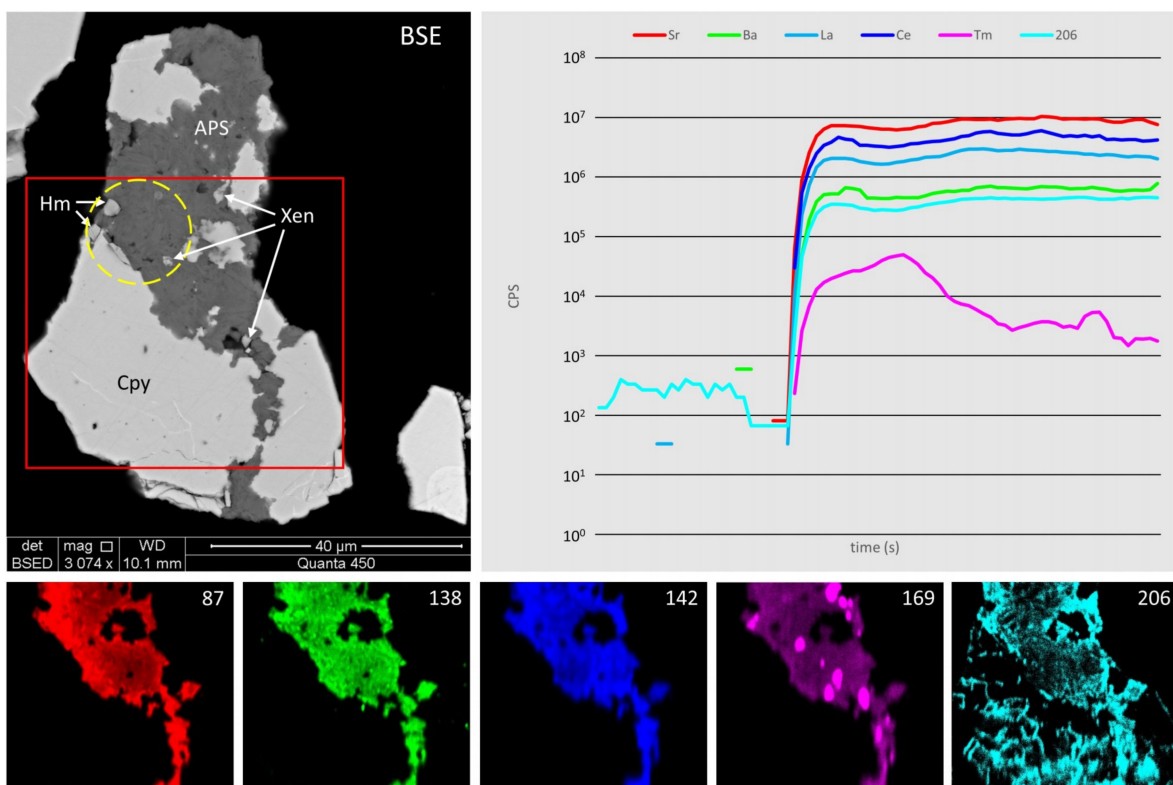

**Figure 8.** The BSE image (top left), single time-resolved laser ablation inductively coupled plasma mass spectrometry (LA-ICP-MS) depth profile data (top right, CPS: counts-per-second), and nanoSIMS images (bottom) of an assemblage of aluminum-phosphate-sulfate (APS) phase(s) with chalcopyrite, hematite, and xenotime in sample 10CLD42. Maps of $^{87}$Sr, $^{138}$Ba, $^{142}$Nd, $^{169}$Tm, and $^{206}$Pb are presented. The red square outline is nanoSIMS mapping area, dashed yellow circle is 15 μm-diameter LA-ICP-MS spot location. Significantly more spatial detail is attainable through nanoSIMS compared to LA-ICP-MS mapping or BSE imaging, although it remains non-quantifiable.

*4.3. Penalty Elements*

Monitoring trace elements may have potential benefits on the other end of the economic spectrum as well. As opposed to payable elements, "penalty elements" may have a negative effect on a saleable product if concentrations are too high. Copper concentrates sold into the global concentrate pool are monitored for concentrations of F, Hg, As, Sb, Bi, Se, Te, and Pb to name a few. Excessive amounts of these may result in penalties, so every effort is made to minimize these in final products. Copper concentrates produced at Olympic Dam are smelted on site, therefore, the spatial locations of smelter deleterious or payable elements within the concentrates are shown here for illustrative purposes only.

Figures 9 and 10 display BSE and nanoSIMS images for selected deleterious elements in different assemblages from four CLD and three FC samples. Arsenic occurs in association with many metals including Fe, Co, Ni, Cu, and Tl, so finding As in pyrite is unsurprising as is the fine oscillatory zoning shown by sample 10FC35. It is more interesting, and less expected, however, that As also appears associated with molybdenite (10CLD24). There are only a few minerals containing both As and Mo, and most of those are in the oxoanionic forms of $AsO_4^{3-}$ and $MoO_4^{2-}$ and are unlikely to be found in sulfide ores. Selenium and Te (also associated with Mo) are frequently found together, both representing group VIa of the periodic table. Molybdenite is known as a suitable trap for various minor minerals, including fine-grained gold and Bi-tellurides, within cleavage domains [52].

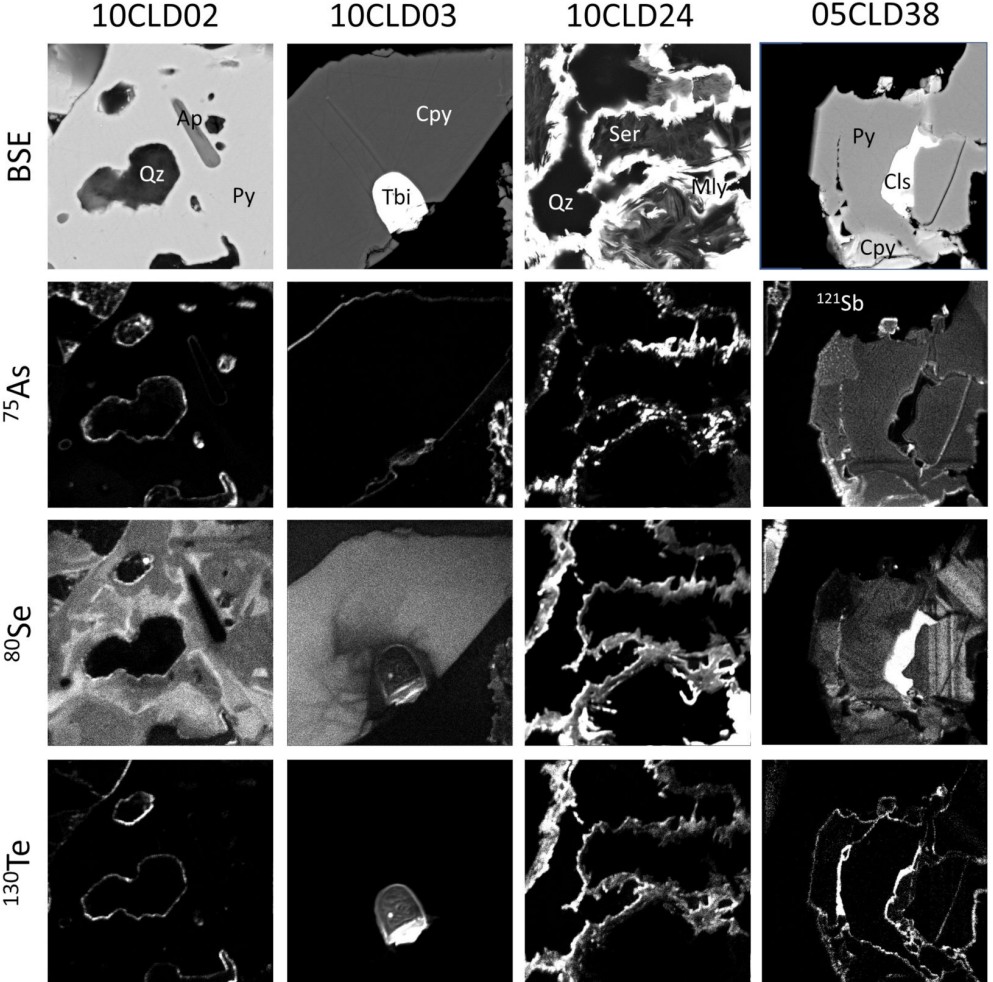

**Figure 9.** The BSE (top row) and nanoSIMS images of potentially deleterious elements ($^{75}$As, $^{80}$Se, $^{121}$Sb, and $^{130}$Te) in different assemblages from the four sulfide concentrate leach discharge (CLD) samples. Abbreviations: Ap-apatite; Cls-clausthalite; Cpy-chalcopyrite; Mly-molybdenite; Py-pyrite; Qz-quartz; Ser-sericite; Tbi-tellurobismuthite. Note that $^{121}$Sb was analyzed instead of $^{75}$As for sample 05CLD38. Previously undetected zoning may be seen with respect to Se in samples 10CLD02, 10CLD03, and 05CLD38.

Commonly associated with Fe, Cu, Ag, Ag, Pd, Bi, and Pb, the chalcogens are found in varying amounts in most pyrite and copper sulfide ore minerals at Olympic Dam. NanoSIMS reveals intricate zoning with respect to Se (10CLD02 and 05CLD38), which is not apparent for Te (or S). Grains 10CLD02 and 05CLD38 illustrate the elusive nature of Te, with high concentrations limited to grain boundaries and microfractures.

One critical penalty element that must be considered during copper concentrate production is Pb. Flotation circuits may be optimized for galena rejection, which solves the bulk of the problem, but Pb in combination with other elements such as Se, Te, Sb, Ag, Cu, or Bi may not behave similarly. Additionally, Olympic Dam ore is well known for extremely fine heterogeneous textures [20], with many minerals occurring as nanosized inclusions (similar to those described in Reference [53]) or fracture fill—also immune to flotation. The deportation of Pb takes on further importance at Olympic Dam where $^{210}$Pb is also a radionuclide. Although low in concentration (only around 13 kg in the entire 10,700 Mt deposit) an activity limit of 1 Bq/g in final concentrates equates to only 380 parts-per-quadrillion (ppq) of $^{210}$Pb. Since targeting $^{210}$Pb specifically by hydrometallurgy is currently not possible, the overall reduction of total Pb in final concentrates is the only reasonable approach to minimizing $^{210}$Pb.

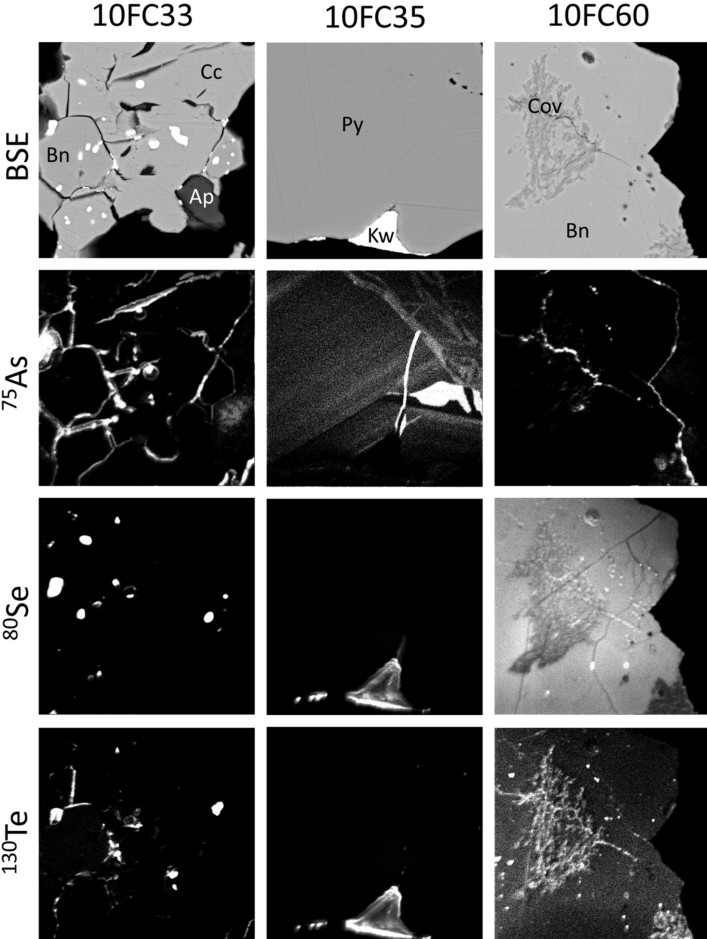

**Figure 10.** The BSE (top row) and nanoSIMS images of potentially deleterious elements ($^{75}$As, $^{80}$Se, and $^{130}$Te) in different assemblages from three sulfide flotation concentrate (FC) samples. Abbreviations: Ap-apatite; Bn-bornite; Cc-chalcocite; Cov-covellite; Kw-kawazulite; Py-pyrite. Zoning with respect to As is evident in sample 10FC35.

In some chalcopyrite and bornite, networks containing elevated concentrations of Pb crosscut individual sulfide grains (Figure 11). These are interpreted as healed microfractures and/or boundaries between sub-grains. Sub-micron zones of a yet-to-be-characterized Pb-bearing component appear at boundaries between minerals, frequently between chalcopyrite and other sulfides. Sample 02FC41 shows such a feature following the mutual boundary between pyrite and chalcopyrite.

Similar features can also be seen on the maps of sample 05CLD85. Sample 02FC44 has more diffuse boundaries, with gradual transitions from chalcopyrite to bornite to chalcocite. In this case, these features appear throughout the bornite, which probably represents an annealed aggregate of small grains rather than a larger single grain, and never extends into adjacent chalcocite or chalcopyrite. Samples 05CLD85 (lower right sector) and 10FC52 show traces of Pb in completely annealed fractures in chalcopyrite, with occasional forays into bornite. During (re-)crystallization, trace amounts of Pb may accumulate in these zones to avoid incorporation into incompatible crystal structures, or the Pb may have been filled in microfractures which have since annealed.

Identifying and characterizing all components within ore-forming hydrothermal fluids may aid in reconstructing ore formation and alteration profiles, and for defining the geochemical signature of the deposit [54]. To the metallurgist, greater interest lies in the ability of nanoSIMS to track trace amounts of potentially economic elements, including Co, Ni, V, In, and REE during various stages of ore processing. Monitoring the fate of these elements after each stage of processing enables metallurgists to classify discrete tailings streams as potential reservoirs for future recovery.

The distributions of potentially deleterious elements are often complex, split between discrete minerals and in solid solution at trace to minor concentrations within multiple host phases. Many of the potential hosts for common elements of concern can be readily predicted on the basis of known geochemical behaviors and published studies from other locations. For example, the relative enrichment of Cd in sphalerite, As in pyrite, or of Bi in bornite are well documented in the literature and supported by quantitative microanalysis (e.g., LA-ICP-MS). Other patterns may, however, be less obvious and can only be picked up by qualitative mapping at appropriate scales.

## 4.4. Radionuclide Distributions

Most IOCG deposits contain some U, with OD being at the higher-concentration end of the spectrum. Fortunately, neither the $^{235}$U and $^{232}$Th decay chains contain any isotopes with the right combination of high activity and a long enough half-life to accumulate in concentrates (months to years); the $^{238}$U chain is more complicated. Efficient removal of $^{238}$U, combined with its long half-life, results in very low activity of $^{238}$U in the final concentrate. The shorter half-lives of a few of the daughter radionuclides, however, create the additional challenge of removing ppb-quantities of these daughters economically, and without a corresponding loss of Cu by co-dissolution. Principal among these daughters is $^{210}$Pb, along with $^{226}$Ra and $^{210}$Po. Our initial hypothesis at the commencement of this project was that by reducing Pb (total) during processing, the $^{210}$Pb concentrations should drop accordingly. This, however, has proven not to be quite as simple as it seemed. NanoSIMS data have confirmed the dissociation between $^{206}$Pb and $^{210}$Pb observed in direct measurements on bulk materials, likely the result of extremely dissimilar residence times.

Six members of the $^{238}$U decay chain can be detected by nanoSIMS; $^{238}$U, $^{234}$U, $^{230}$Th, $^{226}$Ra, $^{210}$Pb, and $^{206}$Pb. A crucial facet of this study is the determination of host minerals for these RNs and any dissociations which may occur. Four examples of RN distributions are presented in Figure 12. All members of the chain are present, to some extent, in most uranium and thorium minerals—especially uraninite (10FC20). Radionuclides in coffinite (10FC06) and brannerite (10FC48) are frequently dissociated from one another at the mineral scale, with variable concentrations and distributions throughout the grains. Although the $^{238}$U in 10FC06 is clearly concentrated in the coffinite, $^{234}$U and $^{230}$Th appear approximately evenly distributed between the coffinite and the surrounding xenotime. Lead-210 and $^{226}$Ra are almost exclusively found in xenotime, and $^{206}$Pb is primarily found as tiny blebs of galena in the center of the coffinite. The $^{206}$Pb-$^{210}$Pb dissociation can be seen in 10FC06, 02FC40, and to a lesser extent in 10FC48.

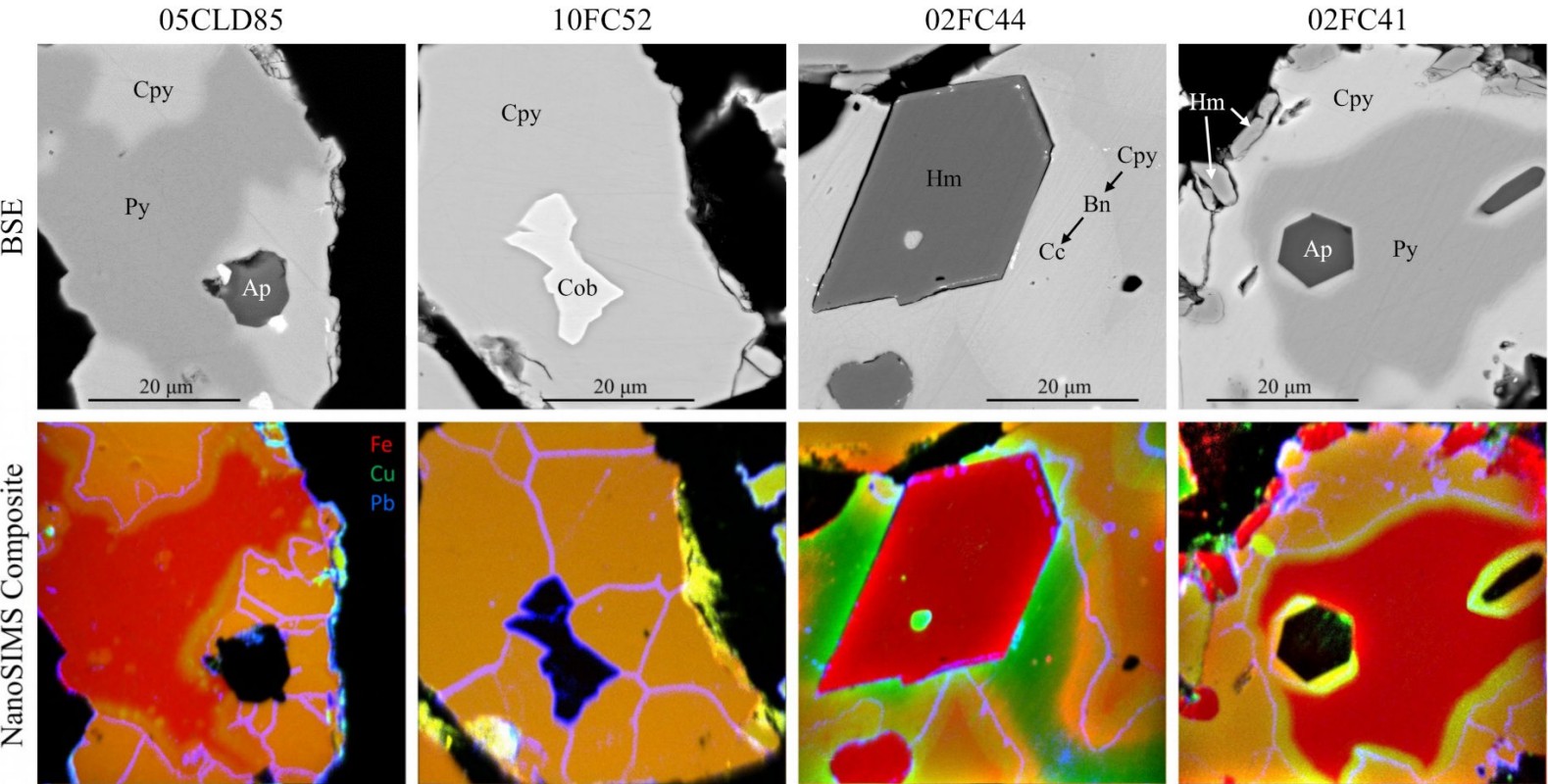

**Figure 11.** The BSE image (upper row) and nanoSIMS composite images of sulfide assemblages in samples 05CLD85, 10FC52, 02FC44, and 02FC41. Abbreviations: Cpy-chalcopyrite; Py-pyrite; Ap-apatite; Cob-cobaltite; Hm-hematite; Bn-bornite; Cc-chalcocite. The composites represent overlapping nanoSIMS maps of Fe (red), Cu (green), and Pb (blue)—the color scheme is uniform for all four composites. Note that orange is the result of a red and green overlap and magenta is the result of a blue and orange overlap. Lead-206 lines/spots are pervasive throughout chalcopyrite and bornite, but are virtually absent in hematite, chalcocite, and pyrite.

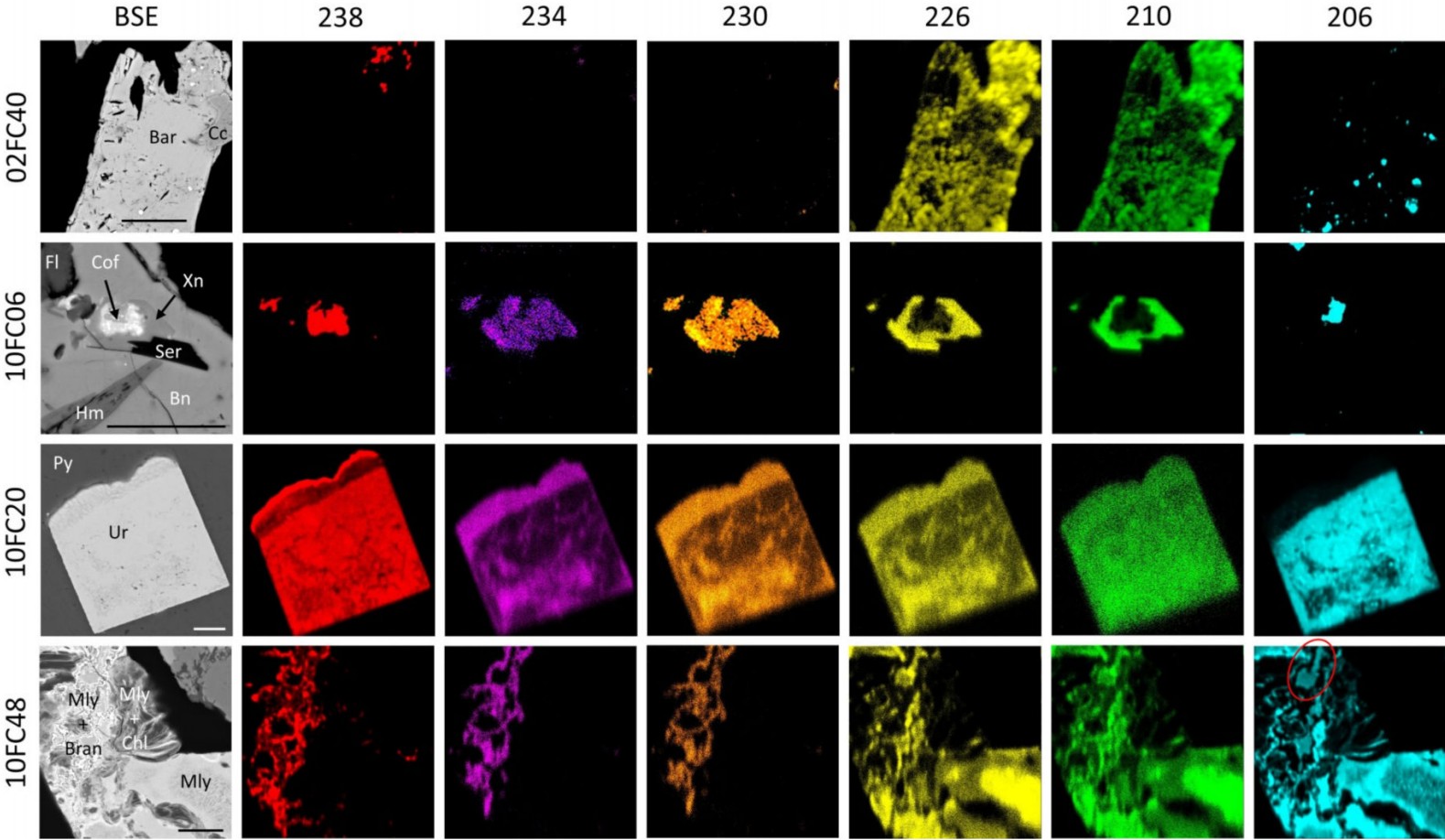

**Figure 12.** The BSE (left column) and nanoSIMS maps of members of the $^{238}$U decay chain from selected areas in samples 02FC40, 10FC06, 10FC20, and 10FC48. Bar-baryte; Bn-bornite; Bran-brannerite; Chl-chlorite; Cof-coffinite; Fl-fluorite; Hm-hematite; Mly-molybdenite; Py-pyrite; Ser-sericite; Ur-uraninite; Xn-xenotime. Scale bars represent 10 μm. Retention of daughter isotopes is strong in uranium minerals (notably uraninite in 10FC20), but dissociation may be prevalent in other minerals without essential U. In sample 10FC48, note the "quarter note" (crotchet) shape in the red oval on the 206 map. Although a complicated pattern, it can be seen that the maps for masses 206, 210, and 226 are in fact the *inverse* of the 230, 234, and 238 maps. In Olympic Dam copper concentrates, the $^{230}$Th/$^{226}$Ra decay transition appears to be the most common point of dissociation amongst RNs.

Baryte (Figure 12, 02FC40; Figure 13) and other sulfate-bearing minerals (e.g., APS phases) are excellent repositories for $^{210}$Pb and $^{226}$Ra, but much less so for RNs from further up the chain. Here, sulfate activity is the controlling factor since both PbSO$_4$ and RaSO$_4$ are extremely insoluble in both water and acids, and baryte is well known as a scavenger of Ra [55–57]. Molybdenite (in sample 10FC48) also seems to only host RN from $^{226}$Ra down the chain, with very little Th or U present. Enrichment of Ra and $^{210}$Pb in the entire molybdenite grain, with higher concentrations in areas of platy textures (lower left) suggest that high surface area cleavage planes may act as a mechanical trap for sulfate-insoluble RNs. The fact that U and Th seem to be immune to this indicates the possibility that nanoparticles of sulfate precipitates are the true host of Ra and Pb and are simply being filtered out of the solution or are adhering to the extensive surface area of molybdenite cleavage planes via electrokinetics.

There are many forms and mineralization stages of baryte at Olympic Dam [29]. Two such types are "dirty" baryte containing a high density of inclusions, and clean, largely homogeneous baryte—both visible in Figure 13. Near the chalcocite/chlorite vein, dirty baryte predominates. Concentrations of $^{226}$Ra and $^{210}$Pb are elevated, and although $^{230}$Th appears to correlate with these, it actually appears to fill boundaries around the chalcocite/chlorite grains. Both U isotopes are confined to the chalcocite/chlorite vein and correlate with REE. Not surprisingly, $^{206}$Pb occurs to some extent everywhere in the mapped area, both coinciding with $^{238}$U and occurring as tiny inclusions (likely galena) within dirty baryte. Interestingly, the "clean" baryte (lowest center-to-left portions of the nanoSIMS maps) is nearly devoid of RNs and $^{206}$Pb. This difference would suggest either two stages of baryte growth or the corrosion/replacement of a pristine baryte crystal by coupled dissolution-reprecipitation processes. The semi-porous nature of the corroded baryte would suggest the latter [58].

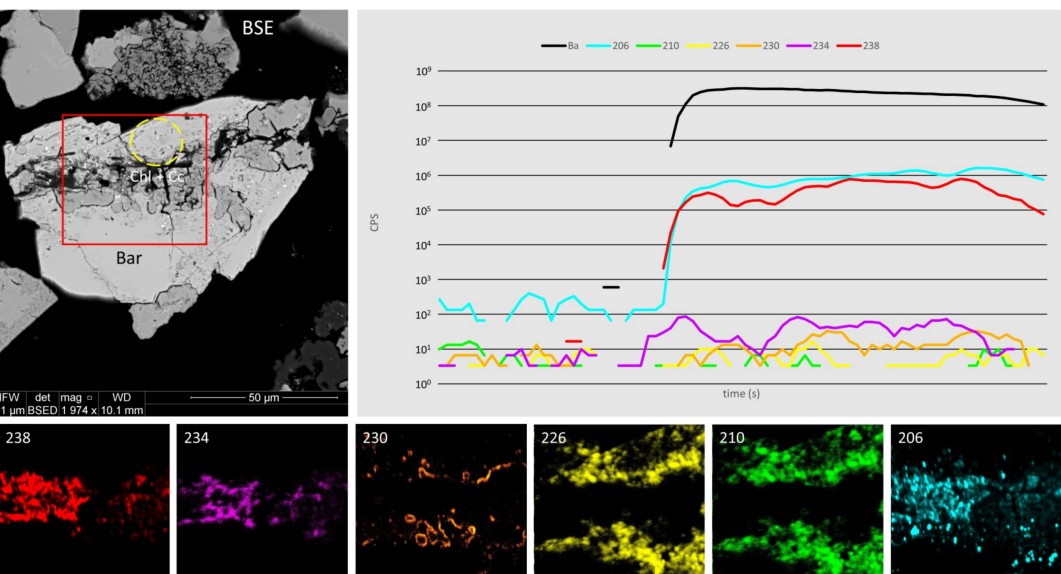

**Figure 13.** The BSE image (top left), single time-resolved LA-ICP-MS depth profile data (top right), and nanoSIMS images (bottom) of assemblage in sample 02CLD29 (chalcocite/chlorite 'vein' crosscutting baryte (Bar) with abundant inclusions). Maps of $^{206}$Pb, $^{210}$Pb, $^{226}$Ra, $^{230}$Th, $^{234}$U, and $^{238}$U are presented. The red outline is the nanoSIMS mapping area, the yellow circle represents the location of the 15 μm-diameter LA-ICP-MS spot. Uranium is concentrated within the vein, whereas $^{210}$Pb, Ra, and Th appear preferentially enriched in baryte containing abundant, dense inclusions. Lead-206 likely has multiple origins and is found throughout. Pristine baryte (lower left of the nanoSIMS maps) contains virtually no RNs.

As opposed to baryte, APS minerals do not seem to display the decoupling of the $^{238}$U chain at the Th-Ra boundary, although there is some considerable variability in the grains observed (e.g., Figure 14). Most RNs seem to reside in small inclusions of xenotime within the APS. What is worthy of note,

however, is the consistent background of all RNs throughout the APS. Although not an overly enriched host, this family of minerals represents approximately 0.1 wt.% of the deposit [20]. When considering concentration limits of certain RNs in the parts-per-trillion range, their relative significance increases.

Exploring the distributions of radionuclides in various minerals from different assemblages from multiple samples raises several questions. Diverse distributions of Pb, Ra, Th, and U may be rationalized by disparities in the chemical behaviors of these elements during fluid-assisted remobilization, but why the apparent dissociation between $^{238}$U and $^{234}$U, and $^{210}$Pb and $^{206}$Pb? Isotopic fractionation will be present, of course, but cannot explain the complete isotopic decoupling of Pb or U. The answers lie at the ends of the decay chain, with the stability of $^{206}$Pb and the exceptionally long half-life of $^{238}$U. Lead-206 is stable and has been produced at Olympic Dam over the past 1.6 billion years, adding to the $^{206}$Pb already present in precursor rocks and ore-forming fluids. It has had sufficient time and exposure to tectonothermal events to mobilize and remobilize multiple times, eventually forming galena or other Pb-chalcogenides [53] and attaining the stability necessary to remain relatively static for millions of years. Lead-210, on the other hand, has a half-life of only 138 days. It is effectively a slave to Ra chemistry since the ingrowth of $^{210}$Pb through the $^{226}$Ra-$^{222}$Rn-$^{218}$Po-$^{214}$Pb-$^{214}$Bi-$^{214}$Po-$^{210}$Pb chain only takes a few days. There is insufficient time for $^{210}$Pb to form Pb minerals as such, though it may augment existing minerals if proximity, time, conditions and growth kinetics allow for it. It is significantly more likely that $^{226}$Ra (half-life ~1600 years) finds stability in accommodating minerals such as sulfates [29], and the $^{210}$Pb simply decays in situ. Although logical, this dissociation has disappointingly nullified attempts to use (readily measurable) $^{206}$Pb as a proxy for (parts-per-trillion-level or less) $^{210}$Pb. Fortunately, however, the nanoSIMS platform eliminates the need for such a proxy.

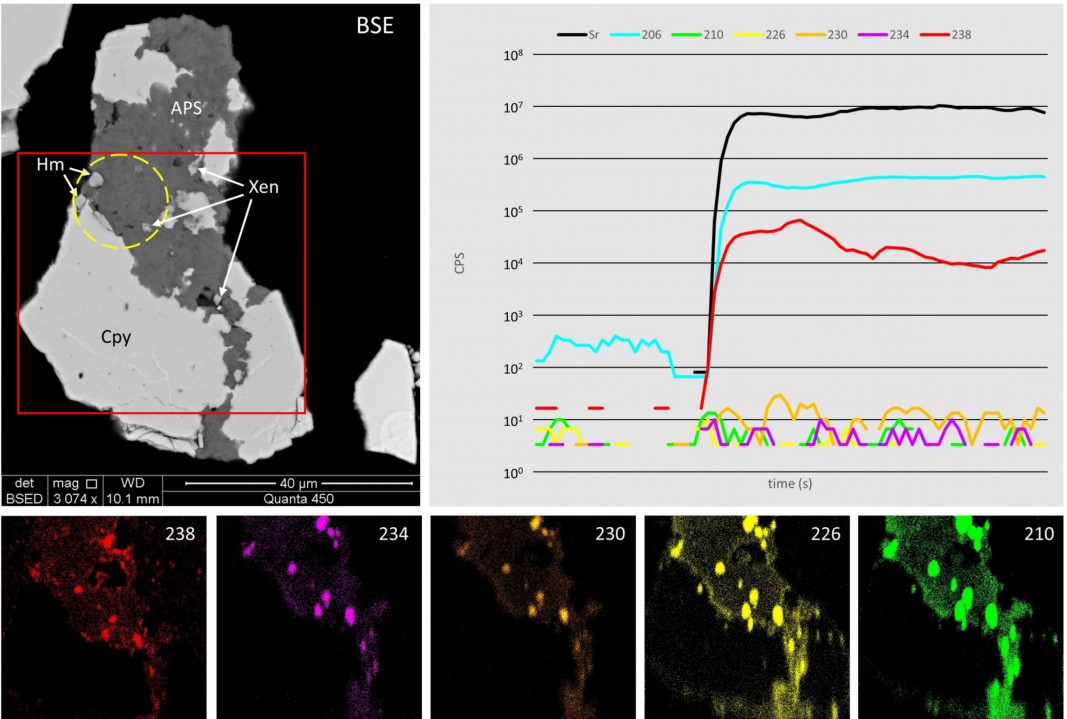

**Figure 14.** The BSE image (top left), single time-resolved LA-ICP-MS depth profile data (top right), and nanoSIMS images (bottom) of an assemblage of APS phase(s) with chalcopyrite (Cpy), hematite (Hm), and xenotime (Xen) in sample 10CLD42. Maps of $^{238}$U, $^{234}$U, $^{230}$Th, $^{226}$Ra, and $^{210}$Pb are presented. The red outline is the nanoSIMS mapping area, the yellow circle is the location of a 15 μm-diameter LA-ICP-MS spot. Only $^{238}$U gives a convincing response by LA-ICP-MS, but all five RNs are clearly present on the nanoSIMS maps with elevated backgrounds in the APS phases and xenotime but not in chalcopyrite.

Uranium pseudo-fractionation is a slightly different mechanism by which fluids containing mostly $^{238}$U (possibly naturally fractionated) infiltrate and begin replacing a crystal already containing the $^{238}$U decay chain in secular equilibrium. The late-stage coffinite replacement of xenotime in 10FC06 (Figure 12), for example, has artificially enriched the grain in $^{238}$U. The top crust of $^{238}$U enrichment on the uraninite grain in 10FC20 is likely the result of a similar mechanism. In short, these apparent "fractionations" between $^{234}$U and $^{238}$U are the result of multiple sources of U in different combinations, rather than the result of a linear chemical process favoring lighter or heavier mass isotopes.

### 4.5. Instrument Capabilities and Future Development

Laser ablation ICP-MS is a remarkably powerful tool which has proven invaluable for detailed characterization of a range of minerals [59]. It has found wide application in mineral characterization of Olympic Dam ores, for hematite [30,32], apatite [35], feldspars [60], rare earth minerals [33,34], and baryte [29], among others. Moreover, microbeam techniques have proven invaluable for the development of new methodologies for dating oxide-dominant ores [61] from Olympic Dam. There are instances, however, when the spatial resolution of LA-ICP-MS is simply inadequate, or where detection limits are not low enough for a given application.

The laser ablation data in Figures 8, 13 and 14 serve to highlight the primary comparative advantage of nanoSIMS for in situ detection of trace elements and isotopes in very fine-grained mineral samples. Lead-206 and $^{238}$U are readily measured and quantified by LA-ICP-MS, but the $^{210}$RN, $^{226}$Ra, $^{230}$Th, and $^{234}$U signals are all barely above the background. It is clear from the laser spot size that even if the signals for these four isotopes were large enough to quantify, all spatial information would be lost.

Equal-scale representations of a 15 μm single time-resolved LA-ICP-MS spot (2500 μm$^3$) and a 50 × 50 × 1 μm nanoSIMS map (also 2500 μm$^3$) clearly illustrate the resolution advantage of nanoSIMS (Figure 15). Granted, though laser ablation spots may be as little as 3 μm in diameter, this dramatically reduces the analyzed volume and usually results in trace elements – especially those with lighter masses–falling below minimum detection limits. A 15 μm-diameter spot size provides higher signals, gives more depth information and is quantifiable (given appropriate matrix-matched standards) but the spatial resolution is clearly limited. For ultra-trace elements, spot diameters of 50 or even 100 μm are not unusual. The nanoSIMS mapped area in the figure assumes an effective spot size of 700 nm, resulting in roughly a 71 × 71 spot grid (pictured). This actually represents the *lowest* resolution for the images above (at 250 pA); running at lower currents can reduce the effective spot size to below 100 nm, resulting in a similar-sized mapped area containing 500 × 500 spots. Ablation depth is estimated at ~1 μm but may be significantly less. The actual depth of ablation is unknown, but it is clear from the BSE images of samples taken after nanoSIMS analyses that different minerals do ablate at different rates.

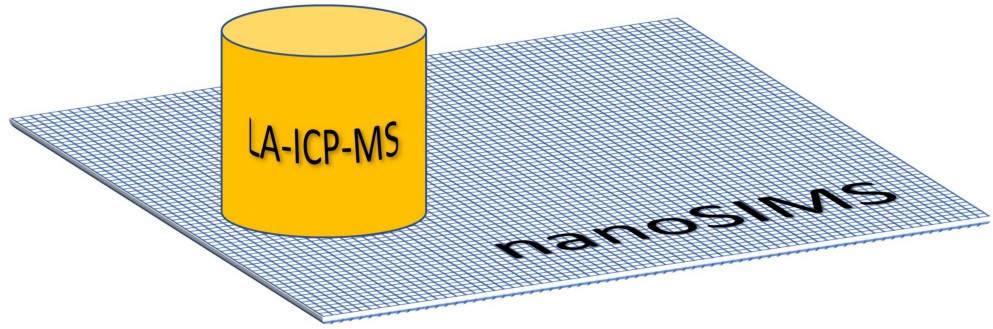

**Figure 15.** The equal-volume representations of a 15 μm single time-resolved LA-ICP-MS crater and a nanoSIMS 50x50x1 μm mapped area. Scales and analysis volumes are equivalent.

As for the greatest shortcoming of nanoSIMS—quantification—we believe this to be both a non-critical and a temporary deficiency. Measuring concentrations of trace elements and isotopes

is always of great interest, but this is impossibly complicated by textures which routinely extend below the micron-scale. LA-ICP-MS spot analysis can provide precise, spatially-resolved concentration data [59], but only as an average of the laser spot volume (at least a few thousands of cubic microns). Judging from the extremely detailed mineral textures presented here, this would be informative yet nevertheless, still a bulk analysis.

The unique information provided by nanoSIMS lies in its spatial resolution without loss of sensitivity, allowing researchers to determine the host minerals—or the lack thereof, wherever boundary domains or surface adherence play a significant role—for trace elements and isotopes in the sub-micron range. Once located, fragments can be extracted in situ for subsequent nanoscale study using techniques such as atom probe tomography [62], or transmission electron microscopy [63], to characterize the materials down to the Ångstrom-scale. Such an approach has been necessary to adequately capture the nanoscale heterogeneity present in several mineral groups from Olympic Dam [28,64–66].

Regarding the longer-term potential for quantification of nanoSIMS analysis, isotope ratios have been measured routinely for over 15 years [67] and relative sensitivity factors have been successfully measured for most elements within uniform matrices [68]. As expected, minerals have proven to be more difficult to quantify, a problem compounded by grain-scale heterogeneity in many natural minerals. Initial tests by the first author have nevertheless shown promising results, with consistent response intensities observed for selected major and minor elements.

## 5. Conclusions

(1) The combined sensitivity and resolution of the nanoSIMS analytical platform provides invaluable information regarding the deportment of trace elements and isotopes and the ultimate destination of components that have been remobilized or reconcentrated, both within the orebody and during mineral processing.

(2) Precious metal extraction may benefit from identifying previously unknown host minerals.

(3) Potentially economic elements may be monitored and archived for future extraction if commodity prices become favorable or game-changing processing methods are developed.

(4) Deleterious "penalty" elements may be closely monitored and reduction/elimination techniques optimized based on new information, especially with respect to hitherto unrecognized host minerals.

(5) Radionuclide daughters of the $^{238}$U decay chain may be located in situ within ores and concentrates and this information used to develop models of their behavior during copper ore processing that will be useful for optimization of extraction.

(6) The nanoSIMS has proven a valuable tool in determining the spatial distribution of trace elements and isotopes in fine-grained copper ores. It complements observations and qualitative data obtained from other methods and offers unique insights into element/isotope distributions in cases where these are too low in absolute concentration and/or too fine to be resolved by other methods. NanoSIMS data can provide researchers with crucial evidence needed to answer questions of ore formation and ore processing.

**Author Contributions:** M.R. and P.G. performed the work presented here. M.R. processed results and made interpretations, supervised by N.J.C. K.J.E. supplied sample material and, together with C.L.C., provided extensive advice. M.R. wrote the manuscript assisted by N.J.C. and all other co-authors.

**Funding:** This is a contribution to the ARC Research Hub for Australian Copper-Uranium (Grant IH130200033), co-supported by BHP Olympic Dam.

**Acknowledgments:** Special thanks are extended to Sarah Gilbert (Adelaide Microscopy) for assistance with laser-ablation ICP-MS analyses.

**Conflicts of Interest:** The authors declare no conflict of interest.

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
