# Peer review of "Detection of Trace Elements/Isotopes in Olympic Dam Copper Concentrates by nanoSIMS"

_minerals, doi:10.3390/min9060336_

Round 1
Reviewer 1 Report
Review to minerals-506264: “Detection of Trace Elements/Isotopes in Olympic
Dam Copper Concentrates by nanoSIMS”
This manuscripts investigates the capability of nanoSIMS to evaluate the distribution of minor and trace elements in ore concentrates. The manuscript is well written and structured.
There are only a few minor suggestions from my side before publication (see below).
Line 101: “nanoscale secondary ion mass spectrometer (nanoSIMS)” … this should be moved to line 75, where nanoSIMS occurs the first time in the manuscript.
Line 233: at the end of the line “gold” should be replaced by “Au”. Same is true for line 234.
Author Response
Thanks for the comments! I've fixed the "acronym first instance" issue (and found another, also fixed). The whole "Se, Te, and Bi" vs. "selenium, tellurium and bismuth" thing is more complicated than I thought, so I referred to the following:
Scientific Style and Format (CSE, 2006, p. 135–140) and the ACS Style Guide (Coghill and Garson, 2006, Chapter 10).
Basically, elements are written out if used as adjectives (e.g. copper concentrates, molybdenum ore), except when hyphenated (e.g. U-bearing). Elements are abbreviated if used as nouns (e.g. inclusions of Bi and Ag), except when used at the beginning of a sentence (e.g. Neodymium and Tm concentrations; Lead-210 and 210Po). There is a distinction between the mineral (gold) and the element (Au). Acronyms (once defined) may be used anywhere (e.g. REE, nanoSIMS).
They might not be intuitive, but at least we can say "that's what we were told to do..."
Reviewer 2 Report
I am very happy with the manuscript, with the way of data presentation, and the interpretation and conclusions taken. I would like to see a bit more elaboration on the radionuclides though. I have added some remarks in this respect directly in the manuscript.
There are some minor and more technical issues which I have also commented directly in the manuscript.
The English is correct and concise. All figures are fine and also correct. If the manuscript needs shortening, which I don't think, the last figure could potentially be omitted.

Author Response
Response to Reviewer #2
I am very happy with the manuscript, with the way of data presentation, and the interpretation and conclusions taken. I would like to see a bit more elaboration on the radionuclides though. I have added some remarks in this respect directly in the manuscript. There are some minor and more technical issues which I have also commented directly in the manuscript. The English is correct and concise. All figures are fine and also correct. If the manuscript needs shortening, which I don't think, the last figure could potentially be omitted. Many thanks for your kind words, helpful comments and the time taken to go through the manuscript in detail. The revised version of the manuscript takes all your comments into consideration, with all changes clearly marked. Below, I attempt explanations to points where no changes were made.
Issues concerning use of element symbols and names written out: The whole "Se, Te, and Bi" vs. "selenium, tellurium and bismuth" thing is more complicated than I thought, so I referred to the following: Scientific Style and Format (CSE, 2006, p. 135–140) and the ACS Style Guide (Coghill and Garson, 2006, Chapter 10).
Basically, elements are written out if used as adjectives (e.g. copper concentrates, molybdenum ore), except when hyphenated (e.g. U-bearing). Elements are abbreviated if used as nouns (e.g. inclusions of Bi and Ag), except when used at the beginning of a sentence (e.g. Neodymium and Tm concentrations; Lead-210 and 210Po). There is a distinction between the mineral (gold) and the element (Au). Acronyms (once defined) may be used anywhere (e.g. REE, nanoSIMS). I tried to follow these conventions, even where they looked strange. They might not be intuitive, but at least we can say "that's what we were told to do..."
Line 73: Why not give a short list of these in-situ techniques? We originally had an extensive list of additional references here but decided to remove them as this is not intended as a review paper. There are plenty of reviews in the published literature which contain that type of information.
Line 82: i don't like this term. Analytically there is no such thing as a maximum detection limit so the minimum detection limit makes so sense. Please only use 'detection limit'. There actually are maximum detection limits (although "detection" may not be the best term there) but analyses like these have a distinct lower AND upper measurement limit. Every instrument has the potential of going off-scale if signal intensities are too high, which actually happened numerous times on the nanoSIMS. Signal intensities are adjustable (to some extent) but less so when those adjustments affect all seven detectors equally. It is not possible to measure iron-56 (in Fe-bearing ore) if you also want to measure lead-210, because the mass 56 signal is above the "maximum limit". I do see your point, but I believe that minimum detection limit is a fairly commonly used phrase, understandable to most readers, even if it seems redundant.
Line 185: There are a number of occasions in the text where you mention detection limits. It would be fine to have at hand here some estimates of detection limits for the ion imaging. I am aware of the fact that absolute detection limits cannot be given but it nevertheless would be fine to have some numbers / a feeling whether we talking about ppb or ppt detection limits for the individual masses. A good point and indeed a tricky issue. We purposefully avoided any quantification because we really don't know. There is considerable variability between elements - and matrices. These issues are being addressed in current work and will be prepared for publication in the future. What I can say for now is that we can map 210Pb in uraninite, which, assuming secular equilibrium, should be around 4 ppb. We're still working on verifying this with other quantitative techniques.
Line 259: I do not fully support this statement: Hydrate ratios on pure Pb or even PbS RMs are always < 10-3 under proper analytical conditions. So what you write is correct. On the other hand the hydrate ratios can be far higher on natural composite samples containing 'wet' minerals, i.e. OH-bearing phases like micas and phosphates and/or interstitual OH-bearing fluids. We have seen Pb-hydrate ratios up to 0.1 on such samples using LA-MC-ICP-MS and Cameca 1280 SIMS. Also long term sample degassing may be problematic in this respect. So please somehow relativate this statement. Very true. We were basing our conclusion on looking at multiple samples, and seeing that some lead minerals had a faint 209 shadow whereas others had none. These are all sulphides in well-degassed samples, so OH- should be minimal. We concluded that it was more likely that the variable 209 signal in lead chalcogenides is due to actual bismuth-209, which we would expect to see anyway. A sentence of clarification has been added.
Line 448: I am missing the information why the 235U and 232Th decay chain products are of no interest. Please add some statements in this respect. Clarification has been added. Basically, the other two decay chains don't have any isotopes with half-lives in the "problem zone", short enough to be very low concentration, but long enough to linger in the copper concentrates.
Line 465: This is probably not quite correct. In my opinion the present view is far to simplicistic: 206Pb can have two different origins: a) radiogenic 206Pb from the in-situ decay of 238U. The amount of radiogenic 206Pb is obviously depending on the amount of 238U and the time since system closure (age). b) inherited 206Pb which in parts is primordial and radiogenic, but unsupported, that is not formed by the in-situ decay of 238U. This inherited 206Pb is what you probably have detected in the PbS in the center of the coffinites. For all we know about the U-Th-Pb system in coffinite and xenotime there must be some radiogenic 206Pb, 207Pb, 208Pb in these minerals. So there is a contradiction of you finding only 210Pb in the xenotime and no 206Pb (and 208Pb nota bene, 207Pb probably being below the detection limit). Please check this and put some more stress on the Pb isotope systematics. Yes, you reiterated our point exactly, which is addressed in lines 524-537. The lead-206 we see is mostly remobilised old lead, shifting around the deposit for 1.6 billion years. The lead-206 from lead-210 decay barely registers, and is many orders of magnitude lower. I changed "only" to "primarily" in line 467, but this is really just a matter of degrees. As far as lead isotope systematics, all we can say is that the lead-206 is geochemically independent. We did not measure leads 207 or 208.
Line 480: Yes, but also the decay of 201Pb to 206Pb seems to cause dissociation (see10FC06 and 10FC20). There seems to be a decoupling of the two Pb isotopes and this is very hard to understand, whereas the decoupling/dissociation of the elements (Th and Ra) is geochemically/mineral chemically far easier to understand as you correctly write in the text. Maybe the Pb isotopes 'suffer' from different recoil effects or such alike. Can you please elaborate on this issue in addition to what you state in lines 519 ff. For what concerns the nanoscale mobility of Pb you my want to check the literature on Pb nanosphere formation in silicates by Monika Kusiak et al. and Elizaveta Kovaleva et al. Again, this has to do with the two distinct sources of lead-206. These sources are temporally separated, not spatially separated. The "dissociation" mentioned here refers to breaks in the active decay chain (of which the Th-230/Ra-226 break is the most distinct), whereas the lead-210/lead-206 non-correlation has to do with the vast majority of lead-206 being removed completely from the decay chain over the past billion years. There are trace amounts of lead-206 wherever we see lead-210, so that part of the active chain is still functioning properly.
Line 532: Although correct this does not explain the Pb behaviour in xenotime and coffinite. See my comments above. It would be fine if you could add some mass balance considerations to this discussion. 206Pb must be in secular equilibrium with 201Pb so you can formulate some boundary conditions for potential 206Pb amounts and relate these to the analytical findings. I think this would be a very important addition to the present outcome. Lead-206 will definitely NOT be in secular equilibrium with lead-210. Only the lead-206 from in situ decay will be in equilibrium. Unfortunately, of the 1,000,000+ tons of lead in the deposit, only 18 kg are lead-210. The xenotime and coffinite have trace amounts of lead-206, but do not accumulate lead-206 through crystallisation or diffusion. That tiny bit of 206 is in equilibrium with the 210, but compared to the deposit as a whole, the 206 (total) signal swamps the 206 (decay) signal. In the images presented, we are generally referring to the 206 (total) signal, as this represents roughly 999,999,982 ppb of the lead. Unfortunately, mass balance considerations would only apply to large-scale bulk samples at best, and degrade completely at the micro- and nanoscale.
Line 545: This is a bit philosophical and can easily be checked by also looking at the 235U decay chain. Open system behaviour, this being what you are writing about, can thereby be readily identified and in favorable cases also be quantified. Philosophical, maybe, but, in our opinion, appropriate for the Discussion section. True, LA-ICP-MS may be able to shed some light on this if we can measure 234, 235, and 238, but resolution becomes an issue. To get a measurable 234 signal, the spot size would need to be larger than the grain. With the excellent resolution of nanoSIMS, we have to trade off quantification - and we are left with some degree of speculation. Improvements in instrument capability may yet lead to resolution of these issues in the future.